# Enhanced Photodetection Range from Visible to Shortwave Infrared Light by ReSe_2_/MoTe_2_ van der Waals Heterostructure

**DOI:** 10.3390/nano12152664

**Published:** 2022-08-03

**Authors:** Zhitao Lin, Wenbiao Zhu, Yonghong Zeng, Yiqing Shu, Haiguo Hu, Weicheng Chen, Jianqing Li

**Affiliations:** 1School of Computer Science and Engineering, Macau University of Science and Technology, Avenida Wai Long, Taipa, Macao 999078, China; 1809853gii30001@student.must.edu.mo (Z.L.); 1909853yii30001@student.must.edu.mo (Y.S.); 2Institute of Microscale Optoelectronics, Collaborative Innovation Centre for Optoelectronic Science & Technology, Key Laboratory of Optoelectronic Devices and Systems of Ministry of Education and Guangdong Province, College of Physics and Optoelectronic Engineering, Shenzhen Key Laboratory of Micro-Nano Photonic Information Technology, Guangdong Laboratory of Artificial Intelligence and Digital Economy (SZ), Shenzhen University, Shenzhen 518060, China; 1910454024@email.szu.edu.cn (W.Z.); zengyonghong777@163.com (Y.Z.); huhaiguo916@163.com (H.H.); 3Guangdong-HongKong-Macao Joint Laboratory for Intelligent Micro-Nano Optoelectronic Technology, Foshan University, Foshan 528225, China; chenwch@fosu.edu.cn

**Keywords:** broadband, heterostructure, interlayer transition, shortwave infrared, photodetector

## Abstract

Type II vertical heterojunction is a good solution for long-wavelength light detection. Here, we report a rhenium selenide/molybdenum telluride (n-ReSe_2_/p-MoTe_2_) photodetector for high-performance photodetection in the broadband spectral range of 405–2000 nm. Due to the low Schottky barrier contact of the ReSe_2_/MoTe_2_ heterojunction, the rectification ratio (RR) of ~10^2^ at ±5 V is realized. Besides, the photodetector can obtain maximum responsivity (R = 1.05 A/W) and specific detectivity (D* = 6.66 × 10^11^ Jones) under the illumination of 655 nm incident light. When the incident wavelength is 1550–2000 nm, a photocurrent is generated due to the interlayer transition of carriers. This compact system can provide an opportunity to realize broadband infrared photodetection.

## 1. Introduction

Two-dimensional (2D) materials have gradually become one of the hotspots of current research since the emergence of graphene. Two-dimensional materials have been widely used in photodetectors [1,2], sensors [3,4], solar cells [5,6] and other fields because of their excellent optical, electronic and mechanical properties [7,8]. Transition metal dichalcogenides (TMDs) in particular, as a class of 2D materials, have attracted extensive attention from researchers [9]. TMDs have adjustable band gap from single layer to bulk, and they can convert direct band gap into indirect band gap by adding the number of layers [10], which enables TMD photodetectors to detect light from visible to near-infrared [11]. Moreover, TMDs have higher environmental stability than black phosphorus, which makes them applicable for devices intended to work in harsh conditions for a long time.

As a kind of typical 2D TMD, rhenium selenide (ReSe_2_) possesses a stable distorted 1T phase, which leads to its in-plane anisotropy [12,13,14]. ReSe_2_ exists as a layer-dependent band structure in the range of 1.3–1.1 eV from monolayer to bulk [15,16,17], which makes it an ideal candidate for constructing high-performance photodetectors in a broad spectrum [18]. Besides, the strong in-plane anisotropic properties of ReSe_2_ have also attracted extensive attention for developing high-performance polarized photodetectors [19].

However, the intrinsic bandgap in ReSe_2_ limits the detection wavelength to near-infrared range [19]. In order to overcome this limitation, heterojunctions based on four different schemes have been proposed [20]. The first scheme uses two materials to form a photoconductive structure, one for absorbing long wavelengths of light and another as a carrier transport channel [21]. The second scheme uses two kinds of materials to form a type II vertical heterojunction structure and uses the characteristics of type II heterojunction to accomplish the carrier interlayer transition [22]. The third scheme is to build a p-g-n structure [23], which uses graphene as a light-absorbing material in the middle layer. Due to the existence of a built-in electric field in the structure, the photogenerated carriers in graphene are effectively separated. The last one is to make a metal array on the surface of a 2D material [24]. It uses the surface plasmon resonance between the incident light and the metal to trap incident light. 

Nowadays, ReSe_2_ has been widely used in photodetectors. Yang et al. constructed the layer-dependent ReSe_2_ photodetector by mechanical exfoliation method, and obtained a photoresponsivity of 95 A/W [17]. The ReSe_2_ doped by Co ions has been discussed by Khan et al., which not only improved the photocurrent of the photodetector, but also broadened the photodetection range [25]. However, the photoresponse speeds of these two photodetectors are relatively slow. Many photodetectors based on the structure of vertical heterojunctions can solve this problem [26]. The InSe/ReSe_2_ heterojunction photodetector was studied by Du et al. and the response speed was 0.36/0.39 ms [27]. However, the photodetection range of ReSe_2_-based vertical heterojunction photodetectors is mostly concentrated in the visible (VIS) to near-infrared band [28]. According to reports, the longest detection wavelength of ReSe_2_-based vertical heterojunction photodetectors is only 1550 nm [29].

In this work, we designed an n-ReSe2/p-MoTe2 vertical heterojunction photodetector for broadband photodetection. Benefiting from the characteristics of type II band alignment, this heterostructure can broaden the photodetection range of our photodetector, which covers visible to shortwave infrared (SWIR) bands. The key parameters of the photodetector were obtained, such as response time, responsivity (R) and specific detectivity (D*). Finally, the energy band alignments of the heterojunction photodetector were analyzed, and the mechanism of carrier interlayer transition was analyzed.

## 2. Results and Discussion

Figure 1a shows a configuration diagram of the ReSe_2_/MoTe_2_ Van der Waals heterostructure. Firstly, the ReSe_2_ and MoTe_2_ nanosheets were stacked on a 275 nm SiO_2_/Si substrate, and then the metal electrodes (Au 60 nm/Cr 10 nm) were deposited onto the two materials by electron beam lithography and electron beam evaporation (shown in experimental details). The optical image of the ReSe_2_/MoTe_2_ heterojunction is shown in Figure 1b. The ReSe_2_ flake is stacked on the MoTe_2_ flake, and metal electrodes are attached to the surfaces of both materials. The thicknesses of ReSe_2_ and MoTe_2_ nanosheets are 15.1 nm and 8.4 nm, respectively (Figure 1c). Additionally, the atomic force microscope (AFM) image of the device is shown in the inset. Besides, the red line in the inset represents the area where the thickness of the heterostructure is obtained. Raman scattering measurements were performed on the heterojunction photodetector using 532 nm laser excitation (Figure 1d). The peaks of 125.1 cm^−1^ and 159.9 cm^−1^ correspond to the E_g_-like and A_g_-like vibration mode of ReSe_2_, respectively [30]. Besides, the Raman characteristic peaks of MoTe_2_ are 173.2 cm^−1^, 233.6 cm^−1^ and 289.8 cm^−1^, which correspond to A_1g_, $E_{2g}^{1}$ and $B_{2g}^{1}$ vibration modes, respectively [31]. Raman characteristic peaks of ReSe_2_ and MoTe_2_ can be observed simultaneously in the heterojunction region. This means that the heterojunction region can reflect the characteristics of the two materials at the same time.

Next, the measurements of photoelectric properties were completed under ambient conditions at room temperature. In the experiment, the drain electrode and the source electrode were contacted to MoTe_2_ and ReSe_2_, respectively. Figure 2a depicts the current-voltage (I–V) characteristics of ReSe_2_/MoTe_2_ in a dark environment. According to the I–V curve, typical rectification characteristics are observed and the rectification ratio is about 10^2^ at ±5 V. It indicates that a heterojunction is formed between ReSe_2_ and MoTe_2_, because the Au/Cr electrode provides good ohmic contact with ReSe_2_ and MoTe_2_, respectively (shown in Appendix A). Furthermore, the source-drain current (*I*_ds_) in both ReSe_2_ and MoTe_2_ materials show positive photocurrent response characteristics under illumination. It is noteworthy that good contact resistance and ohmic characteristics are helpful to improve the performance of the device. The transmission characteristic curve (shown in Appendix A) shows that the current increases with the raising of the positive gate voltage for ReSe_2_, showing an n-type behavior. Whereas the current increases with the raising of the negative gate voltage for MoTe_2_, indicating a p-type behavior. The rectification effect is related to the built-in electric field formed at the p-n junction between ReSe_2_ and MoTe_2_. Under positive bias, the built-in electric field formed at the p-n junction is greatly reduced, which means that electrons are easily transferred across layers leading to a significant conduction current. In contrast, under negative bias, the current is turned off because of the enhanced built-in potential. The quantification of photoresponse to various light intensities is an important experiment to determine the photodetection performance of the heterojunction. Therefore, we measured the characteristics of I-V logarithmic images of the photodetector under different light intensities of 1064 nm laser from dark to 78.1 mW/cm^2^ (Figure 2b). Obviously, under the negative bias, the photocurrent grows with the increase of light intensity, resulting in the reduction of rectification ratio. In addition, the time-dependent photocurrent of the photodetector at −1 V bias was studied under different light intensities of 1.4, 4.8, 15.8, 31.4 and 78.1 mW/cm^2^, which is shown in Figure 2c. As the light intensity was adjusted from 1.4 to 78.1 mW/cm^2^, the photocurrent increased from 1.16 to 9.59 nA. The reason for the high sensitivity under negative bias is that the increase of photoexcited carriers leads to the monotonic increase in photocurrent at higher light intensity. Time-dependent photoresponse of the ReSe_2_/MoTe_2_ heterojunction under varied light intensities (1064 nm, *V*_ds_ = 1 V) has been added in Appendix A shown in Appendix A. Under positive bias, the photocurrent under different light intensities is relatively low and the noise is large. Besides, the photocurrent comparison curves of the photodetector under positive and negative bias are given in Appendix A shown in Appendix A. Next, the photovoltaic characteristics of the photodetector based on the ReSe_2_/MoTe_2_ heterostructure were further investigated in the light intensity range from 1.4 to 63.0 mW/cm^2^, which is shown in Figure 2d. It is worth noting that the photodetector possesses photovoltaic characteristics under light illumination without external bias. The extracted open-circuit voltage (*V*_oc_, left axis) and short-circuit current (*I*_sc_, right axis) show a growth trend (Figure 2e). The maximum values of *V*_oc_ and *I*_sc_ are 0.37 V and 1.18 nA, respectively. At the same time, the corresponding R and D* values under different power densities are calculated and plotted in Figure 2f. These two main parameters are calculated according to the following formula: ${R=(I}_{p}-I_{d}{)/P}_{\mathrm{opt}}$, $D*=R/\sqrt{{2\mathrm{eJ}}_{d}}$, where I_p_, I_d_, P_opt_, e and J_d_ are the photocurrent, dark current, incident light power, electron charge (1.6 × 10^−19^ C) and effective dark current density (J_d_ = 6.3 × 10^−6^ A/cm^2^). The maximum values of R and D* are 485.6 mA/W and 3.4 × 10^11^ Jones (Jones = cm Hz^1/2^ W^−1^), respectively, under the incident light with the power density of 1.4 mW/cm^2^ at −1 V bias. Meanwhile, it can be clearly found that R and D* decrease gradually with the increase of the power density, which is related to the trapping and recombination of the photo-carriers within the heterojunction [32,33,34]. 

The current of the photodetector under negative bias increases obviously (Figure 3a) with the illumination of different wavelengths (405, 532, 655, 808 and 1064 nm). The corresponding linear scale figure has been given in Appendix A. *V*_oc_ and *I*_sc_ under different wavelengths of incident light are extracted and given in the Appendix A (shown in Appendix A). Subsequently, the real-time photoresponse characteristics of the device under laser light of various wavelengths at −1 V are represented in Figure 3b. Since the photoresponse of the device under 1550 nm and 2000 nm incident light is not of the same order of magnitude, the values of the photocurrent in the figure are multiplied by 1000. The phenomenon shown in the figure is not that there will be higher photocurrent under the irradiation of short wavelength incident light. This mainly depends on which wavelength the heterojunction material is sensitive to. Moreover, stable, fast and repeatable photoresponse curves under the laser light of different wavelengths are also given in the Appendix A (shown in Appendix A). The results show that the assembled heterojunction exhibits broadband photosensitivity in the range of ultraviolet (405 nm) to shortwave infrared (2000 nm). In order to further investigate the performance of the ReSe_2_/MoTe_2_ heterostructure photodetector, Figure 3c and Appendix A (shown in Appendix A) illustrate the optical power density dependence of photocurrent at different wavelengths (VIS-SWIR). Under 405–1064 nm light illumination, the photocurrent at *V*_ds_ = −1 V increases monotonously with the raise of laser power at the low optical power density, and reaches saturation at high power density. The non-uniformity of photocurrent dependent on optical power density may be related to the increase of recombination activity of photogenerated charge carriers at high power density and the occurrence of trap states between the Fermi level and conduction band [35]. Under 1550 nm and 2000 nm illumination, the increase of photocurrent does not saturate with the increase of power density. In addition, the wavelength correlation R and D* of the ReSe_2_/MoTe_2_ heterostructure photodetector were also investigated, as shown in Figure 3d. Obviously, the device shows broadband photoresponse in the range of 405–2000 nm, and the peak response of R and D* reaches 1.05 A/W and 6.66 × 10^11^ Jones at 655 nm, respectively. Our heterojunction device can detect visible to shortwave infrared light, even though its response drops dramatically in the 1550–2000 nm region. The R and D* of each wavelength at different power are given in the Appendix A (shown in Appendix A). The responsivity decreases with the increase of incident light intensity. When the incident light intensity increases, more photogenerated carriers will be generated. However, the increase of photogenerated carriers may increase the probability of carriers being trapped by defects, as well as carrier recombination and carrier scattering in the heterojunction, resulting in more carriers being wasted. According to the formula of responsivity, it follows that the more photoelectrons are wasted, the less the photocurrent can be generated, which ultimately leads to the decrease of responsivity.

Figure 4a shows the band alignment of ReSe_2_ and MoTe_2_ before contact. The electron affinities of ReSe_2_ and MoTe_2_ are 4.08 and 3.9 eV, respectively. The band gaps of ReSe_2_ and MoTe_2_ nanosheets are 1.2 and 1.02 eV, respectively [36,37]. According to Appendix A, the electrical characteristics of single ReSe_2_ and MoTe_2_ field-effect transistors (FET) are characterized to determine the carrier type. The ReSe_2_ device exhibits n-type characteristics, while the MoTe_2_ device exhibits p-type characteristics. This means that the Fermi level (E_f_) of ReSe_2_ is close to the conduction band (E_c_), while the Fermi level of MoTe_2_ is close to the valence band (E_v_). After contact, the p-n heterojunction will be constructed, and the transmission of carriers leads to the local band bending at the contact interface. Moreover, the E_f_ will change to the same level because of the equilibrium of carrier transport at the contact interface. Figure 4b shows the band alignment of the heterojunction after contact. The ReSe_2_/MoTe_2_ heterostructure shows the type II staggered band alignment. Due to the difference between the Fermi levels of ReSe_2_ and MoTe_2_, the electrons in ReSe_2_ tend to transfer to MoTe_2_, while the holes in MoTe_2_ transfer in the reverse direction. Figure 4c is the band alignment of the heterojunction under light illumination with a negative bias. Under 405–1064 nm light illumination, photoinduced electrons will be generated in ReSe_2_ and MoTe_2_, respectively. Under the external negative bias, the direction of the built-in electric field of the ReSe_2_/MoTe_2_ heterojunction is consistent with that of the external electric field. Moreover, due to the characteristics of the type II band alignment, the photogenerated electrons in MoTe_2_ flow into ReSe_2_, while the holes in ReSe_2_ flow into MoTe_2_. The resulting separated holes and electrons reside in two different material sheets, and thus photogenerated current is effectively generated. However, under 1550–2000 nm light, since the photon energy is less than the band gap of the two materials, the two materials cannot generate photogenerated carriers alone (Figure 4d). This means that the energy of the incident light is not enough to transfer the electrons in the valence band to the conduction band. Nevertheless, due to the type II band alignment, the electrons in the valence band of MoTe_2_ are able to transfer to the conduction band of ReSe_2_ when the heterojunction is illuminated by long-wavelength incident light, resulting in the interlayer transition. Thus, electrons and holes are effectively separated to form photogenerated current. This is the reason for the excitation of photogenerated current under long wavelengths in Figure 3b.

Figure 5a shows the photoresponse of the photodetector under repeated on/off illumination of 50 Hz, and the photoresponse is normalized. Apparently, our device exhibits outstanding switching characteristics, with fast response speed and excellent reproducibility. Additionally, it has obvious high voltage and low voltage states. Figure 5b shows an enlarged photoresponse curve. According to the definition of response time, the rise and fall times under −1 V bias are estimated to be 5.6 and 4.2 ms, respectively. Appendix A is added to compare the performance parameters of our work and other photodetectors. It can be found that heterojunction photodetectors based on TMD materials can rarely respond to the incident light of shortwave infrared, which is the highlight of our work.

## 3. Conclusions

In this work, we reported an n-ReSe_2_/p-MoTe_2_ vertical heterojunction broadband photodetector. The photodetector exhibits good optoelectronic characteristics from visible to shortwave infrared light (405–2000 nm), where the RR is about 10^2^ at ±5 V. In the photoresponse of the broad spectrum, the maximum R and D* can be obtained under the irradiation of 655 nm incident light, which are 1.05 A/W and 6.66 × 10^11^ Jones, respectively. With the increase of incident light wavelength, the values of R and D* decrease sharply. The analysis of band alignments demonstrates that the photoresponse caused by the interlayer transition is not as strong as that with direct transition. Additionally, the photoresponse speeds of this photodetector are 5.6/4.2 ms. This work also provides reliable evidence for TMD photodetectors for long wavelength photodetection.

## 4. Experimental Details

### 4.1. Preparation of the ReSe_2_/MoTe_2_ Heterostructure Photodetector

ReSe_2_ and MoTe_2_ bulk materials were purchased from Shanghai Onway Technology Co., Ltd., Shanghai, China. Firstly, ReSe_2_ and MoTe_2_ nanosheets were prepared by mechanically exfoliating the bulk materials onto the polydimethylsiloxane (PDMS) stamp. A MoTe_2_ flake was then directionally transferred to a 275 nm thickness SiO_2_/Si substrate. Subsequently, a ReSe_2_ flake was transferred to the substrate to form the heterojunction. Afterwards, the electrode patterns were designed by electron beam lithography (Raith PIONEER Two, Dortmund, Germany). Finally, the source and drain electrodes of 50 nm Au/10 nm Cr were deposited onto the two materials by metal electron beam evaporation technology (Silicon Acer Technology Co., Ltd. ASB-EPI-C6, Zhubei, Taiwan).

### 4.2. Characterization

Raman spectra with 532 nm excitation wavelength were measured by using a Horiba LabRAM (HR800), Paris, France; the laser spot size is about 800 nm. AFM images were collected by Bruker Innovato, Karlsruhe, Germany in order to confirm the thickness of the nanosheets. The electrical and optical performances of the device were tested using a Keithley 4200 semiconductor characteristic analyzer system (Keithley, 4200 SCS), Beaverton, OR, USA under ambient conditions.

## Figures and Tables

**Figure 1 nanomaterials-12-02664-f001:**
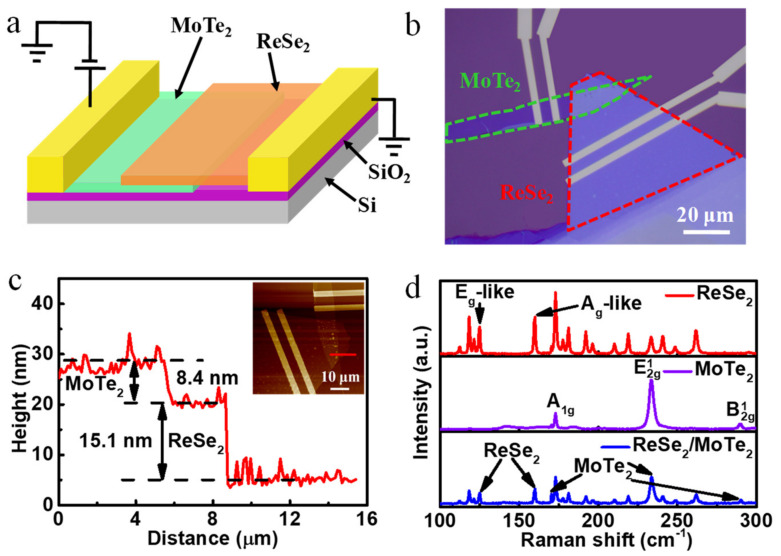
Broadband photovoltaic detector based on p-n structure. (**a**) Configuration diagram of the ReSe_2_/MoTe_2_ heterostructure; (**b**) optical image of a fabricated device; (**c**) individual thickness profiles of ReSe_2_ and MoTe_2_ nanosheets. A topographic AFM image of the ReSe_2_/MoTe_2_ heterostructure is shown in the inset; (**d**) Raman spectra of multi-layer ReSe_2_ and MoTe_2_ nanosheets, as well as the overlapped area in the heterostructure under 532 nm laser.

**Figure 2 nanomaterials-12-02664-f002:**
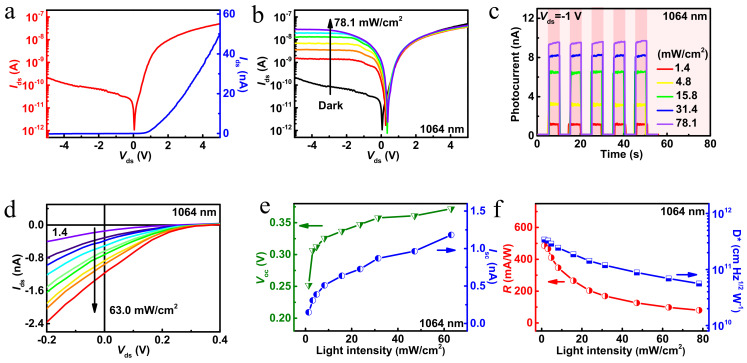
(**a**) The linear and logarithmic coordinates *I*−*V* curves of the heterojunction device in the dark; (**b**) *I*−*V* curves of the ReSe_2_/MoTe_2_ heterojunction under varied light intensities (1064 nm, *V*_ds_ = −1 V); (**c**) time-dependent photoresponse of the ReSe_2_/MoTe_2_ heterojunction under varied light intensities (1064 nm, *V*_ds_ = −1 V); (**d**) enlarged *I*−*V* curves in linear with photovoltaic behaviors; (**e**) extracted open-circuit voltage (*V*_oc_, left axis) and short-circuit current (*I*_sc_, right axis) as functions of light intensity under 1064 nm wavelength; (**f**) responsivity and specific detectivity of the ReSe_2_/MoTe_2_ heterojunction photodetector as functions of light intensity under 1064 nm wavelength.

**Figure 3 nanomaterials-12-02664-f003:**
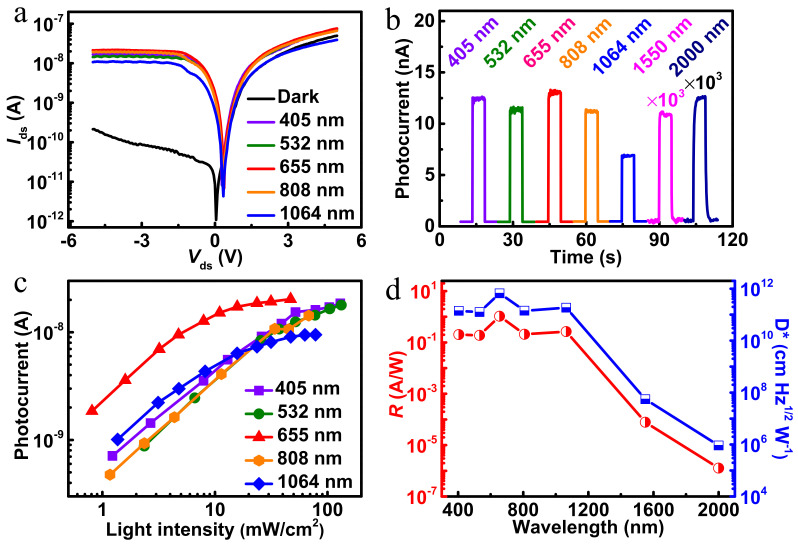
(**a**) *I*−*V* curves of the photodetector under light illumination with different wavelengths; (**b**) temporal photoresponse under various wavelengths (405–2000 nm) with different optical power densities (405 nm @ 39.1 mW/cm^2^, 532 nm @ 37.2 mW/cm^2^, 655 nm @ 7.9 mW/cm^2^, 808 nm @ 34.0 mW/cm^2^, 1064 nm @ 15.8 mW/cm^2^, 1550 nm @ 89.5 mW/cm^2^, 2000 nm @ 6230.9 mW/cm^2^) at −1 V bias, and the values of the photocurrent at 1550 nm and 2000 nm are multiplied by 1000; (**c**) logarithmic plot of the photocurrent as functions of light intensity with different wavelengths; (**d**) responsivity and specific detectivity of the ReSe_2_/MoTe_2_ heterojunction photodetector versus wavelength.

**Figure 4 nanomaterials-12-02664-f004:**
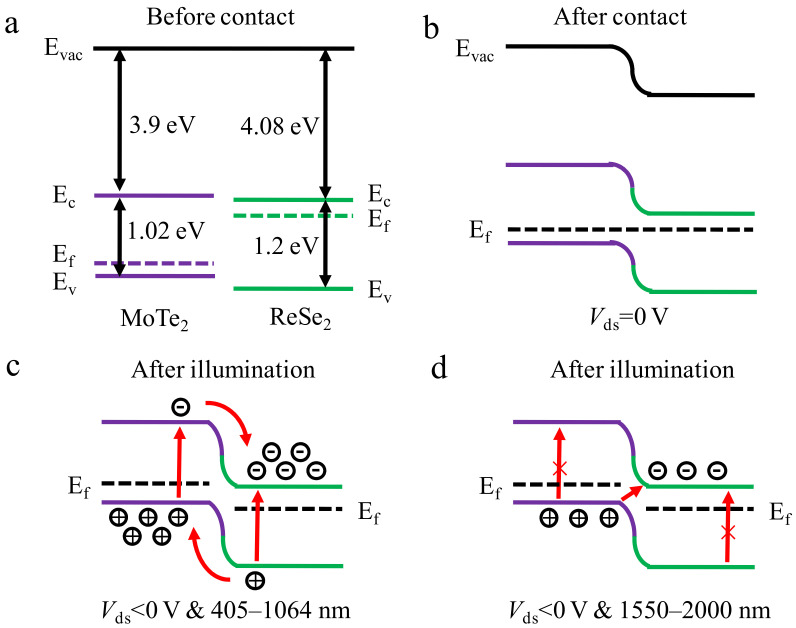
Energy band alignments of the ReSe_2_ and MoTe_2_ before (**a**) and after (**b**) contact. Energy band alignments of the ReSe_2_/MoTe_2_ heterojunction after illumination under negative bias: (**c**) Incident light wavelength is 405–1064 nm. (**d**) Incident light wavelength is 1550–2000 nm.

**Figure 5 nanomaterials-12-02664-f005:**
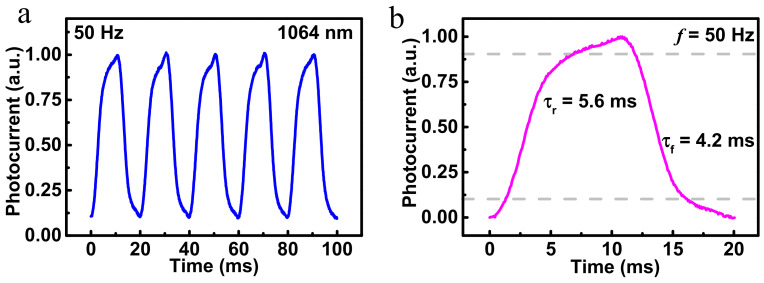
(**a**) Photoresponse of the ReSe_2_/MoTe_2_ heterojunction under 1064 nm light illumination with frequency of 50 Hz; (**b**) rising and falling edges for estimating rise time and the fall time at 50 Hz.

## Data Availability

The data presented in this study are available on request from the corresponding author.

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
