# Peer review of "Enhanced Photodetection Range from Visible to Shortwave Infrared Light by ReSe2/MoTe2 van der Waals Heterostructure"

_nanomaterials, 2022, doi:10.3390/nano12152664_

Round 1
Reviewer 1 Report
In this article the authors have fabricated the type-II heterostructure by MoTe2/ReSe2 and they claimed the enhanced photodetection range from 405 nm to 2000 nm. The work is interesting, but some serious issues should be addressed before publication.
1. What is significance of this work as there are several similar works is already published.
2. What is spot size of laser used in Raman spectroscopy?
3. In Figure 2c, author investigated the photoresponse at Vds= -1V. Please also provide the data at Vds = +1 too.
4. Also, in Figure 3b, the photocurrent at 655 nm is larger than 532 nm. Why?
5. In Figure 3a, IV curves are given in log scale. It would be better if it is provided in linear scale as well.
6. The schematic diagram in Figure 1a could be improve by changing the color contrast.
7. Please mention the thickness of each material with name in Figure 1c.
8. In Figure 4b it must be confirmed that the vacuum level should be bended or not.
9. In Figure 4c and 4d the Vds<0 in both cases of after illumination. It looks strange please check it.
1. In Figure S1, the IV curves are measured but authors did not mention that at which value of ‘Vg’, wavelength and light intensity is used.
1. Why the photoresponsivity is decreasing when intensity is increased as shown in Figure S6?
1. In Figure S3, authors said that they investigated the photo response in atmosphere. Normally, in atmosphere there is oxygen gas and when light is irradiated it make reaction on top surface of 2D materials which ultimately degrade the performance of device. How you controlled this during experiment?
1. It looks strange in Figure S3(a,b) when high wavelengths the photocurrent is also large. Why?
1. Authors claimed that it is innovative Photodetector in conclusion. Since it is not innovative so far. Please change this word.
1. In introduction, authors should discuss other photodetectors to compare with your work. So, it better to discuss more articles in this section like:
(a) DOI https://doi.org/10.1039/D0DT01164A.
(b) https://www.sciencedirect.com/science/article/abs/pii/S0169433221001501
. There are several typos and English must be improved.
Reviewer 2 Report
Authors reported that 2D materials photodetectors using n-ReSe2/p-MoTe2 layers. These photodetectors demonstrated a broad range of superior photodetection from visible (~400nm) to near-infrared (~1000nm). Various analyses and characterization clearly confirmed the detection ability such as responsibility and wavelength defectivity. In addition, these devices showed the possibility of detecting short-wavelength infrared up to 2000 nm. I would like to recommend publishing this manuscript.
Reviewer 3 Report
The authors of this manuscript reported a broadband n-ReSe2/p-MoTe2 photodetector with a spectral range of visible to shortwave infrared. Although the device exhibits good performance, there is a major concern regarding its novelty. A very similar work [Jaffery SHA, et al. "Near-Direct Band Alignment of MoTe2/ReSe2 Type-II p-n Heterojunction for Efficient VNIR Photodetection" Advanced Materials Technologies,2200026 (2022)] has been published recently, which compromises the novelty of this work. This work was not cited in the submitted manuscript and hence no explanation of potential advantages of the new device compared to the previously one is reported. The device reported in the published paper by Jaffery et al. employs the same materials, characterization, and fabrication techniques and the device performance appears to be higher than the new device in the submitted manuscript. Therefore, I cannot recommend the submitted manuscript for publication in Nanomaterials.
Round 2
Reviewer 1 Report
Authors have replied all questions but their responses are not quite satisfied. I would not recommend this manuscript in current form. Following issues must resolved before publication:
1. Authors did not highlighted the the significance of the work well in introduction part.
2. Authors did not response scientifically why photo response is negligible at Vds=+1V. It needs to explain by schematic diagram. I would request to show data in time-resolved measurements not just photocurrent values.
3. It is common that the more sensitive the material is to incident light, the higher the photocurrent. Author should respond according to their configuration of device structures. They should explain why the photocurrent at 655 nm is larger than 532 nm?
4. Usually, the IV curves of ReSe2 and MoTe2 are non-linear with Cr/Au contacts. There are several articles published so far. But you got ohmic contacts (linear IV curves). Why please explain?
5. In Figure S8 which is related to the trapping and recombination of the photo-carriers within the heterojunction like Figure 2f. Please reply with scientific point of view. At which point, within the heterojunction? Normally, when you increase the light intensity, the photocurrent increases and ultimately increase the responsivity initially. But after certain value of light intensity the photoresponsivity start to decrease due to scattering. Please explain this and add in revised manuscript.
6. What is level of oxygen in super clean room?
7. The readers normally focused their interest in main manuscript so in introduction part (especially before last paragraph of introduction), authors should discuss other photodetectors too for generic analysis. So, it better to discuss more articles in this section like: (a) DOI https://doi.org/10.1039/D0DT01164A.
(b) https://www.sciencedirect.com/science/article/abs/pii/S0169433221001501.
8. In conclusion, authors used the “photovoltaic characteristics”. It is better to use “optoelectronic characteristics” because most work covers the photodetection properties not photovoltaic.
9. Conclusion should reflect your output results qualitatively.
Reviewer 3 Report
The revised version has been improved and now deserves to be published. The provided table comparing similar works make it clearer the improvement in device performance and it seems that the new device shows better performance. Therefore, the revised manuscript can be accepted.
